# Whole blood transcriptome biomarkers of unruptured intracranial aneurysm

Kerry E. Poppenberg[1,2,3], Lu Li[4], Muhammad Waqas[3], Nikhil Paliwal[1,2], Kaiyu Jiang[5], James N. Jarvis[5,6], Yijun Sun[5,7], Kenneth V. Snyder[1,3,8,9], Elad I. Levy[1,3,8], Adnan H. Siddiqui[1,3,8], John Kolega[1,10], Hui Meng[1,2,3,11], Vincent M. Tutino[1,2,3,10]*

1 Canon Stroke and Vascular Research Center, Buffalo, New York, United States of America, 2 Department of Biomedical Engineering, University at Buffalo, Buffalo, New York, United States of America, 3 Department of Neurosurgery, University at Buffalo, Buffalo, New York, United States of America, 4 Department of Computer Science and Engineering, University at Buffalo, Buffalo, New York, United States of America, 5 Genetics, Genomics, and Bioinformatics Program, University at Buffalo, Buffalo, New York, United States of America, 6 Department of Pediatrics, University at Buffalo, Buffalo, New York, United States of America, 7 Department of Microbiology and Immunology, University at Buffalo, Buffalo, New York, United States of America, 8 Department of Radiology, University at Buffalo, Buffalo, New York, United States of America, 9 Department of Neurology, University at Buffalo, Buffalo, New York, United States of America, 10 Department of Pathology and Anatomical Sciences, University at Buffalo, Buffalo, New York, United States of America, 11 Department of Mechanical & Aerospace Engineering, University at Buffalo, Buffalo, New York, United States of America

* vincentt@buffalo.edu

## Abstract

### Background

The rupture of an intracranial aneurysm (IA) causes devastating subarachnoid hemorrhages, yet most IAs remain undiscovered until they rupture. Recently, we found an IA RNA expression signature of circulating neutrophils, and used transcriptome data to build predictive models for unruptured IAs. In this study, we evaluate the feasibility of using whole blood transcriptomes to predict the presence of unruptured IAs.

### Methods

We subjected RNA from peripheral whole blood of 67 patients (34 with unruptured IA, 33 without IA) to next-generation RNA sequencing. Model genes were identified using the least absolute shrinkage and selection operator (LASSO) in a random training cohort (n = 47). These genes were used to train a Gaussian Support Vector Machine (gSVM) model to distinguish patients with IA. The model was applied to an independent testing cohort (n = 20) to evaluate performance by receiver operating characteristic (ROC) curve. Gene ontology and pathway analyses investigated the underlying biology of the model genes.

### Results

We identified 18 genes that could distinguish IA patients in a training cohort with 85% accuracy. This SVM model also had 85% accuracy in the testing cohort, with an area under the ROC curve of 0.91. Bioinformatics reflected activation and recruitment of leukocytes,

**Data Availability Statement:** Raw next-generation RNA sequencing data files and tables of processed transcript expression levels for the 47 samples in the training cohort in this publication can be found

at NCBI's GEO (accession no. GSE159610). All expression levels for the LASSO-selected classifier transcripts measured in the 20 samples of the testing dataset are presented in their entirety in the Supporting Information file.

**Funding:** This study was supported by Neurovascular Diagnostics, Inc. in the form of a National Science Foundation (https://www.nsf.gov/) grant awarded to VMT (1746694), The Brain Aneurysm Foundation (https://bafound.org/) in the form of funding awarded to VMT, The New York State Center for Advanced Technology in Big Data and Health Sciences (http://www.buffalo.edu/bioinformatics/resources/funding-incentives/ub-cat.html) in the form of funding awarded to VMT, and The Cummings Foundation (https://jameshcummings.com/) in the form of funding awarded to AHS. The funders had no role in study design, data collection and analysis, decision to publish, or preparation of the manuscript.

**Competing interests:** I have read the journal's policy and the authors of this manuscript have the following competing interests: JNJ—Principal Investigator: NIH Grant R01-AR-060604. KVS—Consulting/teaching: Canon Medical Systems Corporation, Penumbra Inc., Medtronic, Jacobs Institute. Co-Founder: Neurovascular Diagnostics, Inc. EIL—Intratech Medical Ltd. NeXtGen Biologics. Principal investigator: Medtronic US SWIFT PRIME Trials. Honoraria–Medtronic. Consultant–Pulsar Vascular. Advisory Board-Stryker, NeXtGen Biologics, MEDX, Cognition Medical. Other financial support—Abbott Vascular for carotid training sessions. AHS—Financial Interest/Investor/Stock Options/Ownership: Amnis Therapeutics, Apama Medical,BlinkTBI, Inc., Buffalo Technology Partners, Inc., Cardinal Health, Cerebrotech Medical Systems, Inc., Claret Medical, Cognition Medical, Endostream Medical, Ltd., Imperative Care, International Medical Distribution Partners, Rebound Therapeutics Corp., Silk Road Medical, StimMed, Synchron, Three Rivers Medical, Inc., Viseon Spine, Inc. Consultant/Advisory Board: Amnis Therapeutics, Boston Scientific, Canon Medical Systems USA, Inc., Cerebrotech Medical Systems, Inc., Cerenovus, Claret Medical, Corindus, Inc., Endostream Medical, Ltd., Guidepoint 15Global Consulting, Imperative Care, Integra, Medtronic, MicroVention, Northwest University—DSMB Chair for HEAT Trial, Penumbra, Rapid Medical,Rebound Therapeutics Corp., Silk Road Medical, StimMed, Stryker, Three Rivers Medical, Inc.,VasSol, W.L. Gore & Associates. National PI/Steering Committees: Cerenovus LARGE Trial and ARISE II Trial, Medtronic SWIFTPRIME and SWIFT DIRECT Trials,

activation of macrophages, and inflammatory response, suggesting that the biomarker captures important processes in IA pathogenesis.

## Conclusions

Circulating whole blood transcriptomes can detect the presence of unruptured IAs. Pending additional testing in larger cohorts, this could serve as a foundation to develop a simple blood-based test to facilitate screening and early detection of IAs.

## Introduction

Intracranial aneurysms (IAs) are pathological outpouchings within cerebrovasculature whose natural history is driven by inflammation [1–3]. Although the rupture of an IA occurs at a rate of approximately 1% per year, the consequences are devastating. Rupture is the main cause of non-traumatic subarachnoid hemorrhage (SAH), which carries high rates of mortality (up to 50%) [4–6]. Early IA detection would enable closer monitoring and preventive treatments, which can drastically reduce the rate of rupture [7, 8]. For instance, one study found that for a 50-year-old man with an IA, the probability of rupture during his remaining lifetime was 22.8%, but can be reduced to 1.6% after surgical clipping or 3.4% after endovascular coiling [7].

Currently, the only way to diagnose IAs is with cerebral imaging such as MR angiography, computed tomography angiography (CTA), or digital subtraction angiography (DSA). But as the vast majority of IAs are asymptomatic, they are mostly detected incidentally. These imaging modalities are generally not suited for regular IA screening due to prohibitively high costs and potential risks, such as invasive complications (allergic reaction, injection site infection, hematoma, death) and radiation exposure (e.g., DSA, CTA) [9]. And while MRI without contrast enhancement is non-invasive, it is unable to accurately detect many IAs and typically requires invasive follow-up DSA. As it stands, the American Stroke Association does not recommend IA screening in the general population by medical imaging because of the high costs [10], though it has been shown to be cost effecitve in those with an IA family history [11, 12]. Therefore, a blood test would provide an inexpensive, rapid, and minimally-invasive screening test for detecting unruptured IAs in a large population. Those who test positive could then plan for diagnostic imaging and preventive maintenance.

In search of blood-based IA biomarkers, we previously studied gene expression profiles in circulating neutrophils. An initial case-controlled study used RNA-seq to profile neutrophils from individuals with and without IAs (confirmed by angiography) and identified an 82-gene signature that was associated with IA [13]. In a follow-up study, we implemented machine learning to test the feasibility of using gene expression profiles to detect unruptured IAs [14]. The classification algorithm we developed achieved a predictive accuracy of 90% and an area under the curve (AUC) of 0.80 in a small validation cohort. Bioinformatics analyses demonstrated that predictive genes were related to neutrophil activation and dysregulated inflammatory responses, which may explain why they distinguished patients with IAs.

While these studies demonstrated that neutrophil transcriptomes can potentially identify patients with IA, a leukocyte-based diagnostic would not be ideal for clinical implementation. In this case, neutrophils must be processed immediately after collection, and the abundance of neutrophil-derived endonucleases [15] makes it difficult to obtain high quality RNA. An assay using whole blood would overcome these challenges, as whole blood RNAs can be rapidly stabilized at room temperature and do not require rigorous extraction procedures. Such an assay

MicroVention FRED Trial & CONFIDENCE Study, MUSC POSITIVE Trial,Penumbra 3D Separator Trial, COMPASS Trial, INVEST Trial. Principal investigator: Cummings Foundation grant. HM—Principal investigator:NIH Grants R01-NS-091075 and R01-NS-064592. Grant support: Canon Medical Systems. Co-founder: Neurovascular Diagnostics, Inc. VMT—Principal investigator: National Science Foundation Award No. 1746694, Brain Aneurysm Foundation grant, and Center for Advanced Technology grant. Co-founder: Neurovascular Diagnostics, Inc. This does not alter our adherence to PLOS ONE policies on sharing data and materials.

could also be run using standard equipment in diagnostic labs. Thus, in this study, we investigated if gene expression differences in whole blood can distinguish individuals with IA from those without IA, and further, if machine learning could use those differences to build an IA prediction model.

## Methods

### Study enrollment

All methods in this study were approved by the University at Buffalo Institutional Review Board (study no. 030–474433). Written informed consent was obtained from all subjects prior to sample collection and the study was carried out in accordance with the approved protocol. Patients at Gates Vascular Institute (Buffalo, NY) receiving cerebral DSA with and without IA diagnosis were enrolled in this study. Indications for DSA include confirmation of IAs detected on noninvasive imaging or follow-up noninvasive imaging of previously-detected IAs for IA group, or to identify presence or absence of vascular disease (i.e. malformations, carotid stenosis) for control group. Patients who consented to participate in this study were over 18 years old, English-speaking, and had not previously been treated for IA. Patients who were pregnant, had a fever ($>100^\circ$F), recently had invasive surgery, were receiving chemotherapy treatments, had autoimmune diseases, or were on immunomodulating drugs, as noted in their medical records, were excluded. Information about patient's history and comorbidities was collected from electronic medical records.

### Whole blood RNA processing

A volume of 2.5 mL of blood was taken from the femoral access sheath and transferred into a PAXgene blood RNA tube (PreAnalytiX, Hombrechtikon, Switzerland). Total RNA was extracted using the PAXgene Blood RNA kit (Qiagen, Venlo, Limburg, Netherlands) according to manufacturer's instructions. Globin mRNA was removed by magnetic-bead capture with the GLOBINclear kit (Ambion, Austin, TX, USA) following manufacturer's instructions. RNA purity and concentration were assessed by absorbance at 260 nm and were measured by the Quant-iT RiboGreen Assay (Invitrogen, Carlsbad, CA) and the Agilent 2100 BioAnalyzer RNA 6000 Pico Chip (Agilent, Las Vegas, NV), respectively.

### RNA sequencing and data analysis

Libraries were prepared using the Illumina TruSeq stranded total RNA gold kit (Illumina, San Diego, CA). Samples underwent 50-cycle single-read sequencing in a HiSeq2500 Illumina system and were demultiplexed with Bcl2Fastq. After sequencing, per-cycle basecall files generated by the Illumina HiSeq2500 were converted to per-read FASTQ files using bcl2fastq version 2.20.0.422 using default parameters. The quality of the sequencing was reviewed using FastQC version 0.11.5. Potential contamination was detected using FastQ Screen version 0.11.1. No adapter sequences were detected, so no trimming was performed. Genomic alignments were performed using HISAT2 version 2.1.0 using default parameters. NCBI reference GRCh38 was used as reference genome and gene annotation set. Sequence alignments were compressed and sorted into binary alignment map files using samtools version 1.3. Mapped reads for genomic features were counted using Subread featureCounts version 1.6.2 using the parameters -s 2 –g gene_id–t exon–Q 60; the annotation file specified with–a was the NCBI GRCh38 reference provided by Illumina's iGenomes. Raw counts were normalized as transcripts per million (TPM), and ComBat in R was used to correct TPM levels for any bias introduced by sequencing on different flow cells [16–18].

Dispersion of control and aneurysm groups was plotted using $\log_{10}$(TPM+1) for all transcripts with a group average>0 for both aneurysm and control groups. Cell composition analysis was performed using open-access CIBERSORT application (version 1.06) with the TPM normalized gene expression values and the provided 6 cell-type leukocyte signatures (B cells, CD8 T cells, CD4 T cells, NK cells, monocytes, neutrophils) [19]. CIBERSORT uses a linear support vector regression to estimate cell proportions. Transcripts with approved HGNC symbol names (n = 22,924) were used in this analysis. We also visualized how transcriptomes separated control and IA samples by hierarchical clustering via hclust in R under default settings using raw counts with sum>0 across all samples.

## Model gene selection

Prior to selecting genes for model building, we randomly divided the whole blood transcriptome dataset into training and testing cohorts, following a 70:30 split for both aneurysm and control groups. Within the training cohort, we reduced the feature space to protein coding transcripts with an average TPM>1. Candidate genes for the predictive model were then identified using Hilbert-Schmidt Independence Criterion Least Absolute Shrinkage and Selection Operator (HSIC LASSO). Using the $l_1$-regularizer, HSIC LASSO finds a combination of genes that consists of non-redundant features with strong dependence on disease status. Principal component analysis (PCA) with the prcomp function in R visualized how the selected transcripts separate IA from control. Statistical significance of differential expression between IA and control groups was tested by independent samples t-test for equal variance and by Mann-Whitney U test for unequal variance.

## Development of IA prediction model

Features selected by LASSO were used to train a classification model by SVM with a Gaussian kernel in MATLAB's Statistics and Machine Learning Toolbox. SVM has been successfully used in a variety of disease classification applications, including our previous efforts using neutrophil transcriptomes [14]. A 10-fold cross-validation within the training set was performed as the model was being developed to reduce likelihood of overfitting. We compared model predictions to clinical diagnoses to determine number of true positives, true negatives, false positives, and false negatives, which were used to calculate sensitivity, specificity, and accuracy as defined elsewhere [14]. Receiver operating characteristic (ROC) curves were created, from which area under ROC curve (AUC) was calculated as a metric of model performance in the training cohort. We also calculated positive and negative predictive values (PPV, NPV) to examine how disease prevalence influences model's predictive ability using equations enumerated elsewhere [14]. The model was subsequently applied to the independent testing cohort. The TPM values of LASSO-selected features for each subject in the testing cohort were input to the model by a blinded operator to make predictions. Then, the true diagnoses of the subjects (positive or negative for IA) were compared to model predictions to evaluate model performance in the testing cohort.

## Bioinformatics

Ontologies that were significantly enriched in model genes compared to the background list of 34,605 expressed genes in our sequencing were identified using the Gene Ontology enRIchment anaLysis and visuaLizAtion tool (GORILLA) [20]. We reported biological processes with a p-value<0.0005. Pathways and networks associated with the LASSO genes were studied with Ingenuity Pathway Analysis (IPA) software [21], using fold-changes calculated in the training cohort for the 18 model genes. IPA maps our genes of interest to a

gene object in the Ingenuity Knowledge Base to create networks based on known interactions between products of the genes and to identify enriched ontologies and upstream regulators. Networks with a p-score ≥15, ontologies that assign at least 3 model genes and have a p-value<0.05, and upstream regulators with an activation z-score≥|1.5| were considered significant.

# Results

## Study population characteristics

We included a total of 67 peripheral blood samples (34 IA, 33 control) that met our inclusion/ exclusion and quality criteria in this study. These samples were randomly divided into an n = 47 training cohort (24 IA) and an n = 20 testing cohort (10 IA). Control and IA populations in both cohorts had similar demographics and comorbidities (Table 1), with the exception of smoking, which was higher in the IA training group. Aneurysm size (largest diameter measured on DSA) ranged from 1 to 19 mm, with a mean of 5.6 mm (S1 Table). There were 41 IAs total, as 6 patients had multiple IAs.

RNA quality and sequencing metrics are reported in S2 Table. The 67 sequenced samples had an average 260/280 of 1.9 and average RNA integrity number of 8.4. On average, 57.29 million sequences per sample and 96% aligned rate were obtained. Expression dispersion between IA and control groups are visualized in Fig 1A. To verify differentially expressed transcripts were derived from expression differences related to presence of IA, rather than differences in cell populations, we estimated the proportions of different cell populations in each sample using CIBERSORT [19]. This showed no statistically significant difference in proportions of cell types between control and IA groups. On average across all samples, neutrophils represent the majority (45%) followed by monocytes (19%), CD4 T cells (16%), CD8 T cells (8%), B cells (7%), and NK cells (5%) (Fig 1B).

All 67 transcriptomes were sorted by supervised hierarchical clustering using raw counts without any feature reduction. Generally, clusters were either predominantly IA or control, as

**Table 1. Clinical characteristics of training and testing cohorts*.**

|  | Training Cohort | | Testing Cohort | |
|---|---|---|---|---|
|  | Control (n = 23) | Aneurysm (n = 24) | Control (n = 10) | Aneurysm (n = 10) |
| **Age (Mean±SE)** | 58±3.6 | 55±2.6 | 56±4.0 | 57±4.6 |
| Over 55 | 60.87% | 45.83% | 40.00% | 60.00% |
| **Sex** |  |  |  |  |
| Female | 56.52% | 70.83% | 60.00% | 90.00% |
| **Smoker** |  |  |  |  |
| Yes | 0.00% | 33.33% | 10.00% | 30.00% |
| **Comorbidities** |  |  |  |  |
| Hypertension | 30.43% | 29.17% | 20.00% | 40.00% |
| Heart Disease | 17.39% | 16.67% | 0.00% | 20.00% |
| High Cholesterol | 34.78% | 33.33% | 40.00% | 30.00% |
| Stroke History | 17.39% | 8.33% | 10.00% | 0.00% |
| Diabetes | 13.04% | 8.33% | 10.00% | 0.00% |
| Osteoarthritis | 26.09% | 25.00% | 30.00% | 30.00% |

*Clinical characteristics of the randomly-created training and testing cohorts. With exception of age, these factors were quantified as binary data points. The clinical factors were retrieved from patients' medical records via latest "Patient Medical History" form administered before imaging. SE = standard error.

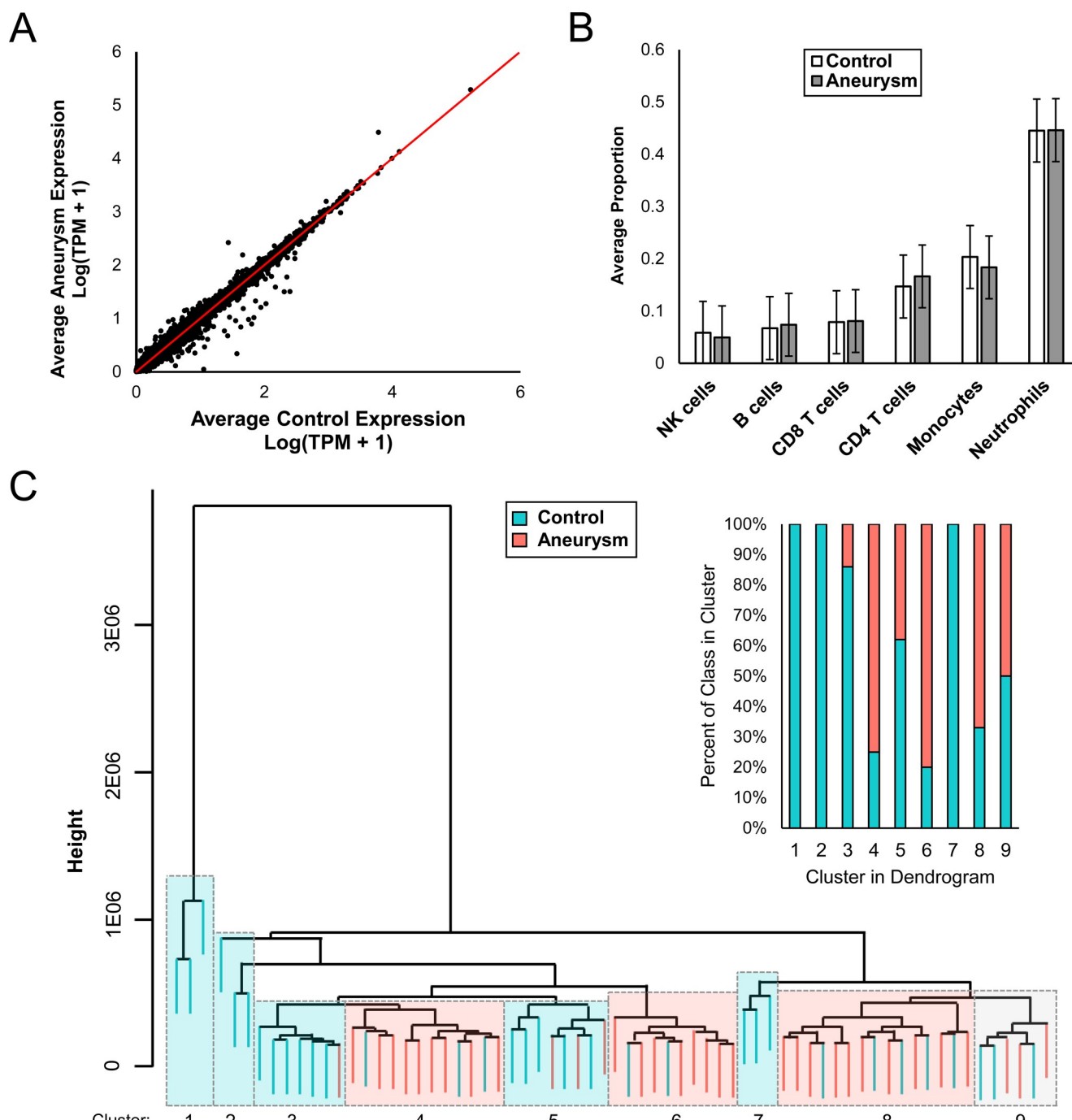

**Fig 1. Differential expression analysis. A)** Scatterplot depicts dispersion in expression between IA and control groups. **B)** No difference between cell type proportions of aneurysm and control groups was found. In both, neutrophils comprise majority of cells expressed in whole blood transcriptomes. **C)** Hierarchical clustering using genes with TPM sum>0 for all 67 whole blood transcriptomes. Teal indicates control samples, while pink indicates aneurysm samples.

seen in Fig 1C. The first few clusters in the dendrogram are composed of mainly control samples, but progressing rightwards, there are larger groups of mostly aneurysm samples followed by a group of equal composition at the far right.

## gSVM model can detect IA with high accuracy

We employed a regression-based technique, LASSO, to select genes with the greatest predictive ability (S3 Table) to use in our model. Fold-change and p-value of the 18 genes in the training cohort are reported in Table 2. PCA (Fig 2A) using these 18 genes demonstrates they are able to clearly separate disease from control cases in the training cohort. The first 3 components capture the majority (56%) of total variance. These 18 genes were used to train our SVM prediction model.

In training, this model achieved greater than 80% accuracy, sensitivity, and specificity. Using an estimated prevalence of 5% [22], we calculated that the model had an NPV of approximately 1 (Fig 2B). ROC analysis confirmed the robust predictive ability as the model had an AUC of 0.92 (Fig 2C). We examined how this set of model genes could separate IA and control groups within the independent testing cohort, consisting of 20 patients (10 IA). PCA visualization demonstrates that these transcripts can still discriminate IA and control groups within this new dataset, again accounting for the majority of variance (64%) within the first 3 principal components (Fig 2D). As shown in Fig 2E and 2F, the model performed well in independent testing with an accuracy of 85% and an AUC of 0.91.

## Genes in model reflect inflammatory processes

Biological functions of the 18 model genes were investigated through gene ontology and pathway analysis. GORILLA identified 4 significant processes: "negative regulation of secretion", "negative regulation of protein secretion", "negative regulation of peptide secretion", and "cytokine-mediated signaling pathway" (Table 3). IPA analysis indicated 2 significant networks (Fig 3A and 3B) with functions related to cell death and survival, cardiovascular system development and function, and tissue development (A); cancer, endocrine system disorders, and gastrointestinal disease (B). Network A is a highly connected network with dense signaling

**Table 2. 18 transcripts selected during model training.**

| Gene | Gene ID | Accession no. | F-C | P-value |
|---|---|---|---|---|
| ATF3 | 467 | NM_001674 | -1.86 | <0.001 |
| CBWD6 | 644019 | NM_001085457 | 1.35 | 0.001 |
| CCDC85B | 11007 | NM_006848 | 1.35 | 0.001 |
| CCR8 | 1237 | NM_005201 | 1.68 | <0.001 |
| CHMP4B | 128866 | NM_176812 | -1.14 | 0.007 |
| CLEC4F | 165530 | NM_173535 | -2.81 | 0.002 |
| CXCL10 | 3627 | NM_001565 | -2.64 | <0.001 |
| FN1 | 2335 | NM_212476 | -2.88 | 0.06 |
| MT2A | 4502 | NM_005953 | -1.65 | <0.001 |
| MZT2B | 80097 | NM_025029 | 1.18 | 0.008 |
| PCSK1N | 27344 | NM_013271 | 1.56 | 0.018 |
| PIM3 | 415116 | NM_001001852 | 1.31 | <0.001 |
| SLC37A3 | 84255 | NM_032295 | 1.23 | 0.032 |
| ST6GALNAC1 | 55808 | NM_018414 | 1.71 | <0.001 |
| TCN2 | 6948 | NM_000355 | -1.77 | <0.001 |
| TIFAB | 497189 | NM_001099221 | -1.48 | 0.007 |
| TNFRSF4 | 7293 | NM_003327 | 1.48 | <0.001 |
| UFSP1 | 402682 | NM_001015072 | 1.32 | 0.003 |

P-values were calculated in the training dataset by independent t-test if equal population variances, Mann-Whitney U test if not. No. = number, F-C = fold-change.

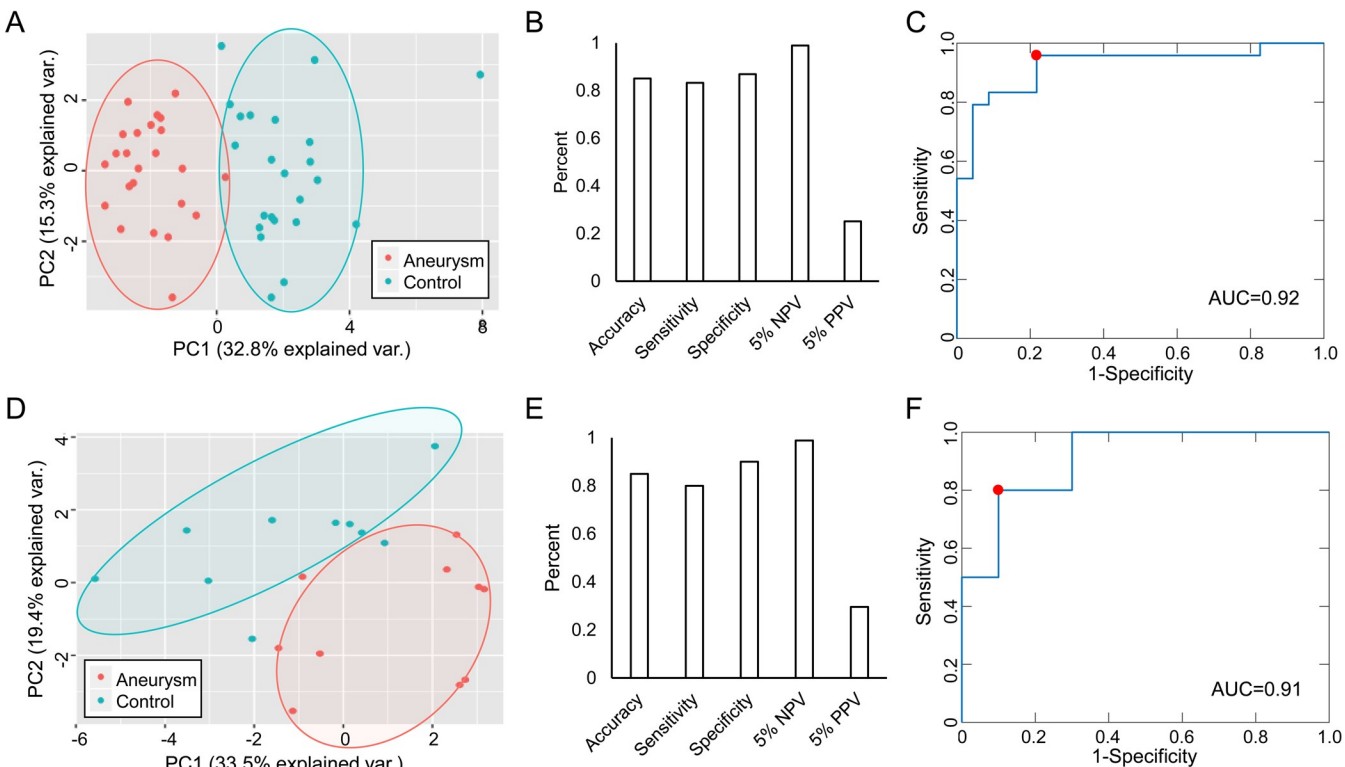

**Fig 2. Performance of 18 gene SVM biomarker in training and testing.** *Top*: Training. **A)** PCA shows this panel can distinguish between aneurysm (red) and control (blue) samples. **B)** Accuracy, sensitivity, specificity, 5% NPV, and 5% PPV of model in training. **C)** ROC curve for model has AUC of 0.92. *Bottom*: Testing. **D)** PCA illustrates panel was able to separate samples in a new cohort. **E)** Accuracy, sensitivity, specificity, 5% NPV, and 5% PPV of model in testing. **F)** ROC shows high performance in testing cohort (AUC = 0.91).

centered around AKT, ERK, FN1, IL1, JNK, MAPK, PI3K, and VEGF. In Network B, TP53 and CTNNB1 function as central nodes. Genes in each network are reported in S4 Table. The disease and biological functions reported by IPA include activation of leukocytes, cell death of immune cells, activation of macrophages, inflammatory response, recruitment of leukocytes, apoptosis of leukocytes (full list presented in S5 Table). Progesterone, OSM, and IL1B (the latter two being important cytokines involved in inflammatory signaling [23, 24]) were upstream regulators predicted to be inhibited (Fig 3D).

## Discussion

There is a critical need for a minimally-invasive prescreen to identify patients who have an unruptured IA and would, therefore, maximally benefit from cerebral vascular imaging (such as MRA) for IA detection. Previously, we hypothesized that circulating blood cells have altered expression profiles after contact with IA tissue or inflammatory mediators released by IAs

**Table 3. GORILLA ontologies for the 18 transcripts selected by LASSO.**

| GO term | Description | P-value | Genes |
|---|---|---|---|
| GO:0051048 | Negative regulation of secretion | 4.29E-05 | FN1, PIM3, TIFAB, TNFRSF4 |
| GO:0050709 | Negative regulation of protein secretion | 2.46E-04 | FN1, PIM3, TNFRSF4 |
| GO:0002792 | Negative regulation of peptide secretion | 2.71E-04 | FN1, PIM3, TNFRSF4 |
| GO:0019221 | Cytokine-mediated signaling pathway | 3.23E-04 | CCR8, CXCL10, FN1, MT2A, TNFRSF4 |

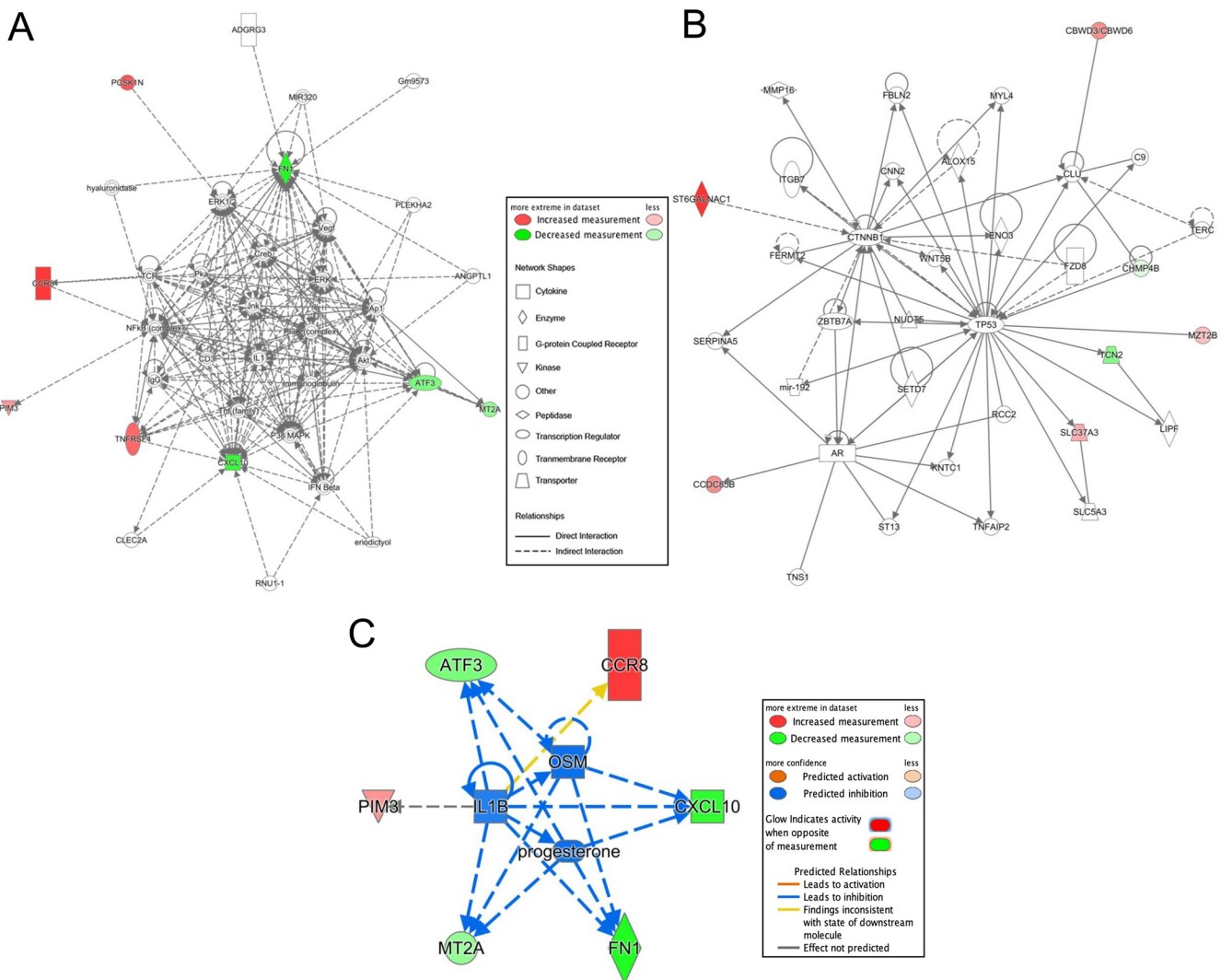

**Fig 3. IPA Network analysis of 18 genes identified by LASSO.** Transcripts with increased expression in IA are red; transcripts with lower expression are green; fold-change represented by intensity. **A)** The first network (p-score = 20) reflects cardiovascular system development and function, cell death and survival, and tissue development. **B)** The second network (p-score = 17) has ontologies of cancer, endocrine system disorders, and gastrointestinal disease. **C)** Network constructed using 3 significant upstream regulators (progesterone, OSM, IL1B).

[25]. We investigated this by performing a transcriptome profiling study of circulating neutrophils in patients with and without IA and found an IA signature in neutrophils [13], which when trained via a machine learning pipeline demonstrated predictive ability to detect unruptured IAs. In this study, we discovered that a unique IA signature exists in whole blood transcriptomes, albeit there were no common genes between neutrophil and whole blood IA biomarkers.

Our data yielded 18 genes that discriminated IA from control cases via LASSO regression. This type of feature selection overcomes shortcomings of traditional statistical filtering, since filtering methods consider genes independently, neglecting functional interactions between genes/gene products. Consequently, feature selection by traditional filtering methods may omit genes that constructively work together during a particular disease state, and may select redundant genes. During training, we used an SVM algorithm which separates binary labeled

samples by transforming them into a multidimensional space and establishing a hyper-plane that maximizes the distance between samples of either class. Our Gaussian SVM (gSVM) model using the 18 selected genes had a prediction accuracy of 85% in both the training cohort (via cross-validation) and an independent testing cohort. Using isolated neutrophils in our previous study, we achieved a maximum accuracy of 90% in an independent testing cohort. The decrease in observed biomarker performance may be due to additional noise associated with a heterogeneous cell population in whole blood. There may also be greater inter-patient variability with whole blood transcriptomes due to contribution of multiple cell types. However, the whole blood model still achieved an NPV>0.98 in both training and testing, supporting its feasibility as a pre-screen for IA, for which high NPV is desired.

We suspect that the 18 classifier transcripts detect IAs because they capture key facets of the disease, related to inflammation, infiltration, and degradation of the IA wall. Four of the model genes (*TNFRSF4*, *TIFAB*, *MT2A*, *PIM3*) are associated with NF-κB, an important inflammatory signaling pathway implicated in IA pathogenesis [26]. Most notably, NF-κB upregulates MMP-9 [27], a main driver of IA wall degradation [28], and MCP-1, which recruits macrophages to the IA wall (a hallmark of aneurysmal tissue) [28]. *TNFRSF4* (increased in our study) is a member of the TNF-receptor superfamily that is involved in NF-κB pathway activation and has also been found to be increased in aneurysm tissue. Both the TNF family and NF-κB complex contribute to vessel degradation and are captured in the first IPA-derived network with direct connections to multiple model genes (*ATF3*, *CCR8*, *CXCL10*, *FN1*, *PIM3*, *TNFRSF4*). TNFα, a cytokine within the TNF family, has increased expression in plasma of aneurysm patients [29] and IA walls, and leads to EC dysfunction, inflammation, and apoptosis [28]. Conversely, *TIFAB* and *MT2A* (both decreased in our study) inhibit activation of NF-κB, suggesting another mechanism for NF-κB activation in patients with IAs.

A role for inflammation is also reflected in other biomarker genes (*CCR8*, *CXCL10*) related to cytokine/chemokine signaling. For example, *CCR8* (increased in our study) is a member of the beta chemokine receptor family and has greater expression in M1 pro-inflammatory macrophages (vs. M2 macrophages). M1 macrophages have been shown to be more prevalent in IAs [30] and may contribute more to pathologic remodeling during IA natural history [31]. Critical signaling pathways are also represented in the first network, with the transcription factor AP1 interacting with ERK, JNK, and MAPK complexes. AP1 has been linked to other inflammatory diseases, such as arthritis [32], and has been shown to regulate MMP-2, which could degrade extracellular matrix in IA [33]. Furthermore, a predicted upstream regulator of AP1 is IL1B, which is significantly increased in plasma of IA patients [29] and is associated with extracellular matrix destruction, NF-κB signaling, and vascular SMC apoptosis [34]. IL1B upregulates adhesion molecules on endothelial cells that recruit neutrophils and monocytes, and can induce both reactive oxygen species production and MMP-9 degradation via NGAL [35, 36]. Another important cytokine implicated in IPA analysis is OSM, an upstream regulator. OSM can regulate production of other cytokines, including IL6, which has been implicated in polymorphism studies of multiple populations [37–39].

While numerous model genes have clear associations to IA pathogenesis via inflammatory and signaling pathways, others are related to functions that have not been extensively explored in IA. For example, intra-and extra-cellular signaling, reflected by *ATF3*, *CHMP4B*, and *PCSK1N* may play a role in the complex reactions of circulating cells to IA presence. Overall, these and the other remaining genes require further study to elucidate their roles in IA pathogenesis, since they may represent unique predictive targets in whole blood RNA expression profiles. One way of determining which transcripts are most associated with IA may be to investigate if their "signal" increases when blood is collected from the intracranial vessels or from the aneurysm sac, as others have done [25, 40]. We hypothesize that differences in the

IA-associated transcripts would be exaggerated in blood samples drawn closer to the IA tissue, as the circulating cells that confer transcriptomic changes due to a blood-IA interaction would be most concentrated at that location. Future studies are required to test this hypothesis.

This study has several limitations. First, this is a single-center study, which may have introduced selection bias into our experimental design. Second, every subject underwent imaging by DSA. Thus, while our control population was confirmed to not have an IA, they may have other health issues that prompted DSA imaging. Furthermore, we used DSA for identification of IAs, as it is the gold standard in cerebrovascular imaging because of its high resolution. However, future studies may use less sensitive modalities, such as CTA or MR angiography, to confirm the presence of IA, which therefore, may result in false positives for the proposed biomarker. Third, there was an imbalance in comorbidities between IA and control groups that could have contributed to differential expression. To address these limitations, we are currently planning a multi-center study to prospectively validate our biomarker in patients receiving both DSA and non-invasive imaging, such as MRA. This large study will increase both our sample size and the diversity of patient population. It will also allow us to incorporate multiple control groups, such as those with other types of aneurysm or vascular abnormalities, to identify transcripts most specific to IA.

## Conclusions

In this study we developed an accurate (85%) machine learning classifier derived from whole blood transcriptomes to predict presence of unruptured IA. Bioinformatics analyses indicate that critical inflammatory pathways are captured by the model genes, which is consistent with our previous findings using neutrophils. While other groups have studied whole blood transcriptomes for IA biomarkers, they used single, small RNA molecules [41, 42], did not perform cerebral imaging on control subjects, or did not use an independent testing cohort. We addressed these shortcomings by confirming presence or absence of IA with cerebral imaging, using gSVM with a panel of genes to better handle inter-sample variability, performing feature identification and model construction in a separate training cohort, and assessing true model performance in an independent testing cohort. While we implemented an improved study design, we still need to confirm our biomarker in a large, multi-center study.

## Supporting information

**S1 Table. Characteristics of 27 aneurysms in all patients with intracranial aneurysms (6 patients had multiple intracranial aneurysms).**
(DOCX)

**S2 Table. Cohort assignment and RNA quality.**
(DOCX)

**S3 Table. Per-gene performance of the 18 model transcripts.**
(DOCX)

**S4 Table. Transcripts in the 2 significant networks constructed by ingenuity pathway analysis (IPA).**
(DOCX)

**S5 Table. Significant disease and biological functions assigned by ingenuity pathway analysis for genes identified by LASSO.**
(DOCX)

**S6 Table. The testing data.**
(DOCX)

## Acknowledgments

We thank the patients who participated in this study, Jonathan Bard MA and Brandon Marzullo MS for RNA sequencing data analysis assistance, and Jennifer L. Gay CCRP for study protocol management. This work was performed in part at the New York State Center of Excellence in Bioinformatics and Life Sciences' Genomics and Bioinformatics Core.

## Author Contributions

**Conceptualization:** Kerry E. Poppenberg, John Kolega, Hui Meng, Vincent M. Tutino.

**Data curation:** Kerry E. Poppenberg, Muhammad Waqas, Kenneth V. Snyder, Elad I. Levy, Adnan H. Siddiqui, Vincent M. Tutino.

**Formal analysis:** Kerry E. Poppenberg, Lu Li, Nikhil Paliwal.

**Funding acquisition:** Vincent M. Tutino.

**Investigation:** Kerry E. Poppenberg.

**Methodology:** Kerry E. Poppenberg, Kaiyu Jiang, Yijun Sun.

**Software:** Lu Li.

**Supervision:** James N. Jarvis, Yijun Sun, Vincent M. Tutino.

**Visualization:** Kerry E. Poppenberg.

**Writing – original draft:** Kerry E. Poppenberg, Vincent M. Tutino.

**Writing – review & editing:** Kerry E. Poppenberg, James N. Jarvis, John Kolega, Hui Meng, Vincent M. Tutino.

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
