## [Decision Letter · Decision Letter 0]

16 Sep 2020

PONE-D-20-25156

Whole blood transcriptome biomarkers of unruptured intracranial aneurysms

PLOS ONE

Dear Dr. Tutino,

Thank you for submitting your manuscript to PLOS ONE. After careful consideration, we feel that it has merit but does not fully meet PLOS ONE’s publication criteria as it currently stands. Therefore, we invite you to submit a revised version of the manuscript that addresses the points raised during the review process.

We look forward to receiving your revised manuscript.

Kind regards,

Jinglu Ai, M.D., Ph.D.

Academic Editor

PLOS ONE

Journal Requirements:

"I have read the journal's policy and the authors of this manuscript have the following competing interests:

JNJ—Principal Investigator: NIH Grant R01-AR-060604.

KVS—Consulting/teaching: Canon Medical Systems Corporation, Penumbra Inc., Medtronic, Jacobs Institute. Co-Founder: Neurovascular Diagnostics, Inc.

EIL—Intratech Medical Ltd. NeXtGen Biologics. Principal investigator: Medtronic US SWIFT PRIME Trials. Honoraria–Medtronic. Consultant–Pulsar Vascular. Advisory Board-Stryker, NeXtGen Biologics, MEDX, Cognition Medical. Other financial support—Abbott Vascular for carotid training sessions.

AHS—Financial Interest/Investor/Stock Options/Ownership: Amnis Therapeutics, Apama Medical,BlinkTBI, Inc., Buffalo Technology Partners, Inc., Cardinal Health, Cerebrotech Medical Systems, Inc., Claret Medical, Cognition Medical, Endostream Medical, Ltd., Imperative Care, International Medical Distribution Partners, Rebound Therapeutics Corp., Silk Road Medical, StimMed, Synchron, Three Rivers Medical, Inc., Viseon Spine, Inc. Consultant/Advisory Board: Amnis Therapeutics, Boston Scientific, Canon Medical Systems USA, Inc., Cerebrotech Medical Systems, Inc., Cerenovus, Claret Medical, Corindus, Inc., Endostream Medical, Ltd., Guidepoint

15Global Consulting, Imperative Care, Integra, Medtronic, MicroVention, Northwest University—DSMB Chair for HEAT Trial, Penumbra, Rapid Medical,Rebound Therapeutics Corp., Silk Road Medical, StimMed, Stryker, Three Rivers Medical, Inc.,VasSol, W.L. Gore & Associates. National PI/Steering Committees: Cerenovus LARGE Trial and ARISE II Trial,Medtronic SWIFTPRIME and SWIFT DIRECT Trials, MicroVention FRED Trial & CONFIDENCE Study, MUSC POSITIVE Trial,Penumbra 3D Separator Trial, COMPASS Trial, INVEST Trial.

HM—Principal investigator:NIH Grants R01-NS-091075 and R01-NS-064592. Grant support: Canon Medical Systems. Co-founder: Neurovascular Diagnostics, Inc.

VMT—Principal investigator: National Science Foundation Award No. 1746694, Brain Aneurysm Foundation grant, Center for Advanced Technology grant, and Cummings Foundation grant. Co-founder:Neurovascular Diagnostics, Inc."

Reviewers' comments:

Reviewer's Responses to Questions

**Comments to the Author**

1. Is the manuscript technically sound, and do the data support the conclusions?

Reviewer #1: Yes

Reviewer #2: Yes

2. Has the statistical analysis been performed appropriately and rigorously? 

Reviewer #1: Yes

Reviewer #2: Yes

3. Have the authors made all data underlying the findings in their manuscript fully available?

Reviewer #1: Yes

Reviewer #2: Yes

4. Is the manuscript presented in an intelligible fashion and written in standard English?

Reviewer #1: Yes

Reviewer #2: Yes

5. Review Comments to the Author

Reviewer #1: The authors present work on a transcriptome method of unruptured IA. This is both a critical public health need and appears to be conducted in a scientifically sound manner which improves on previous work.

I recommend this manuscript for publication. There are a few issues which must be corrected:

1) Proofreading. One example: "Currently, the only way to diagnose IAs is with cerebral imaging such as MR angiographjy, 59 computed tomographgy angiography (CTA), or digital subtraction angriography (DSA)." This is easily solved and does not affect either understanding or deliberation.

2) Please include in the discussion what differences in signal occur testing peripheral blood compared to blood taken from an intracranial catheter during DSA.

3) Mention that DSA is the gold standard and that CTA/MRA is less sensitive and what affect that may have, please.

4) Femoral access is done with a sheath, not a true catheter.

Reviewer #2: This is a well written manuscript & of great interest to all physicians who are involved in caring for subjects with unruptured intracranial aneurysms.

One minor concern/question:

Were the authors blinded when they performed the analysis on the tested group (not controlled/trained)??

If not, then this MUST be done first before establishing the sensitivity, specificity, accuracy, PPV & NPV??

6. PLOS authors have the option to publish the peer review history of their article (what does this mean?). If published, this will include your full peer review and any attached files.

Reviewer #1: No

Reviewer #2: No

---

## [Author Response · Author response to Decision Letter 0]

19 Oct 2020

Authors response to reviewer comments for Manuscript PONE-D-20-25156, entitled “Whole blood transcriptome biomarkers of unruptured intracranial aneurysms”:

Authors: We thank the editors and reviewers for their insightful comments, which have helped to improve the manuscript. On the basis of these comments, we have made several changes, which are tracked by yellow highlighting in the revised manuscript. Below are our point-by-point responses to the comments. The page numbers mentioned correlate with the revised version of the manuscript. Thank you.

Editorial Comments:

Journal Requirements: 

Authors: We have ensured that our manuscript meets PLOS ONE’s style requirements.

2. Thank you for stating the following [not shown] in the Competing Interests section. Please confirm that this does not alter your adherence to all PLOS ONE policies on sharing data and materials, by including the following statement: "This does not alter our adherence to PLOS ONE policies on sharing data and materials.” (as detailed online in our guide for authors http://journals.plos.org/plosone/s/competing-interests). If there are restrictions on sharing of data and/or materials, please state these. Please note that we cannot proceed with consideration of your article until this information has been declared. Please include your updated Competing Interests statement in your cover letter; we will change the online submission form on your behalf.

Authors: The conflicts of interest do not alter our adherence to all PLOS ONE policies on sharing data and materials. We have included the Competing Interests statement in the cover letter and added the following language: 

"This does not alter our adherence to PLOS ONE policies on sharing data and materials.”

3. We note that you have indicated that data from this study are available upon request. PLOS only allows data to be available upon request if there are legal or ethical restrictions on sharing data publicly. In your revised cover letter, please address the following prompts: a) If there are ethical or legal restrictions on sharing a de-identified data set, please explain them in detail (e.g., data contain potentially sensitive information, data are owned by a third-party organization, etc.) and who has imposed them (e.g., an ethics committee). Please also provide contact information for a data access committee, ethics committee, or other institutional body to which data requests may be sent. b) If there are no restrictions, please upload the minimal anonymized data set necessary to replicate your study findings as either Supporting Information files or to a stable, public repository and provide us with the relevant URLs, DOIs, or accession numbers. For a list of acceptable repositories, please see http://journals.plos.org/plosone/s/data-availability#loc-recommended-repositories.

Authors: There are some legal restrictions, as the data are partially owned by a third-party organization, Neurovascular Diagnostics, with whom this study was performed in conjunction. The data can still be accessed via request to their Data Administrator. As per PLOS ONE’s policy, we have now our data availability statement: 

“Data cannot be shared publicly because it is partially owned by the study sponsor, Neurovascular Diagnostics, Inc. Data are available from the Neurovascular Diagnostics Data Access Committee (contact Dr. Hamidreza Rajabzadeh-Oghaz, Data Administrator at info@nvdiag.com, hacademic1811@gmail.com [alternate], or 1-866-552-6402) for researchers who meet the criteria for access to confidential data.”

This statement has been added to the cover letter. We have also added this information to the revised manuscript on page 12, under “Availability of RNA expression data”.

Reviewer Comments:

Reviewer 1:

1. Proofreading. One example: "Currently, the only way to diagnose IAs is with cerebral imaging such as MR angiographjy, 59 computed tomographgy angiography (CTA), or digital subtraction angriography (DSA)." This is easily solved and does not affect either understanding or deliberation.

Authors: Thank you for pointing this out. We have gone through the entire manuscript to correct spelling errors and typos throughout. 

2. Please include in the discussion what differences in signal occur testing peripheral blood compared to blood taken from an intracranial catheter during DSA.

Authors: This is a good suggestion. Intuitively, we suspect that blood closer to the aneurysm may give greater signal in the biomarker. Differences in truly IA-associated transcripts would likely be exaggerated in blood samples drawn closer to the aneurysm tissue because the circulating cells that confer transcriptomic changes due to a blood-IA interaction would be most concentrated there. This is now discussed in the Discussion on page 15 of the revised manuscript. 

3. Mention that DSA is the gold standard and that CTA/MRA is less sensitive and what affect that may have, please.

Authors: Thank you for this suggestion. In the limitations section of the Discussion on page 15 we now have stated that DSA is the gold standard, and that in future studies the use of CTA or MR angiography (which may be less sensitive to IA presence) to diagnose IAs could unintentionally give false positives for the proposed biomarker.

4. Femoral access is done with a sheath, not a true catheter.

Authors: Thank you for this comment. We have now corrected this in the Methods on page 4 of the revised manuscript.

Reviewer 2: 

1. Were the authors blinded when they performed the analysis on the tested group (not controlled/trained)? If not, then this MUST be done first before establishing the sensitivity, specificity, accuracy, PPV & NPV?

Authors: Thank you for this question. Yes. The operators who input the data into the machine learning models were blinded to the class of the subjects. Only after the predictions were made, did the operator check the prediction with the true aneurysm status (yes or no) of the patient. This is now detailed in the Methods on page 7 of the revised manuscript.

---

## [Editor Report · Decision Letter 1]

22 Oct 2020

Whole blood transcriptome biomarkers of unruptured intracranial aneurysms

PONE-D-20-25156R1

Dear Dr. Tutino,

We’re pleased to inform you that your manuscript has been judged scientifically suitable for publication and will be formally accepted for publication once it meets all outstanding technical requirements.

Kind regards,

Jinglu Ai, M.D., Ph.D.

Academic Editor

PLOS ONE
---

## [Editor Report · Acceptance letter]

28 Oct 2020

PONE-D-20-25156R1 

Whole blood transcriptome biomarkers of unruptured intracranial aneurysm 

Dear Dr. Tutino:

I'm pleased to inform you that your manuscript has been deemed suitable for publication in PLOS ONE. Congratulations! Your manuscript is now with our production department. 

Kind regards, 

on behalf of

Dr. Jinglu Ai 

Academic Editor

PLOS ONE